# The Combination of Molnupiravir with Nirmatrelvir or GC376 Has a Synergic Role in the Inhibition of SARS-CoV-2 Replication In Vitro

**DOI:** 10.3390/microorganisms10071475

**Published:** 2022-07-21

**Authors:** Anna Gidari, Samuele Sabbatini, Elisabetta Schiaroli, Sabrina Bastianelli, Sara Pierucci, Chiara Busti, Lucia Comez, Valeria Libera, Antonio Macchiarulo, Alessandro Paciaroni, Ilaria Vicenti, Maurizio Zazzi, Daniela Francisci

**Affiliations:** 1Department of Medicine and Surgery, Clinic of Infectious Diseases, “Santa Maria della Misericordia” Hospital, University of Perugia, 06132 Perugia, Italy; elisabetta.schiaroli@unipg.it (E.S.); sabrina.bastianelli@unipg.it (S.B.); sara.pierucci@unipg.it (S.P.); chiarabusti93@gmail.com (C.B.); daniela.francisci@unipg.it (D.F.); 2Department of Medicine and Surgery, Medical Microbiology Section, University of Perugia, 06132 Perugia, Italy; samuele.sabbatini@unipg.it; 3IOM-CNR, Department of Physics and Geology, University of Perugia, 06123 Perugia, Italy; comez@iom.cnr.it (L.C.); valeria.libera@studenti.unipg.it (V.L.); 4Department of Pharmaceutical Sciences, University of Perugia, 06123 Perugia, Italy; antonio.macchiarulo@unipg.it; 5Department of Physics and Geology, University of Perugia, 06123 Perugia, Italy; alessandro.paciaroni@unipg.it; 6Department of Medical Biotechnologies, University of Siena, 53100 Siena, Italy; ilariavicenti@gmail.com (I.V.); maurizio.zazzi@gmail.com (M.Z.)

**Keywords:** COVID-19, SARS-CoV-2, molnupiravir, nirmatrelvir, GC376, PF-07321332, variant, synergy, antiviral

## Abstract

Introduction: The development of effective vaccines has partially mitigated the trend of the SARS-CoV-2 pandemic; however, the need for orally administered antiviral drugs persists. This study aims to investigate the activity of molnupiravir in combination with nirmatrelvir or GC376 on SARS-CoV-2 to verify the synergistic effect. Methods: The SARS-CoV-2 strains 20A.EU, BA.1 and BA.2 were used to infect Vero E6 in presence of antiviral compounds alone or in combinations using five two-fold serial dilution of compound concentrations ≤EC90. After 48 and 72 h post-infection, viability was performed using MTT reduction assay. Supernatants were collected for plaque-assay titration. All experiments were performed in triplicate, each being repeated at least three times. The synergistic score was calculated using Synergy Finder version 2. Results: All compounds reached micromolar EC90. Molnupiravir and GC376 showed a synergistic activity at 48 h with an HSA score of 19.33 (*p* < 0.0001) and an additive activity at 72 h with an HSA score of 8.61 (*p* < 0.0001). Molnupiravir and nirmatrelvir showed a synergistic activity both at 48 h and 72 h with an HSA score of 14.2 (*p* = 0.01) and 13.08 (*p* < 0.0001), respectively. Conclusion: Molnupiravir associated with one of the two protease-inhibitors nirmatrelvir and GC376 showed good additive-synergic activity in vitro.

## 1. Introduction

The coronavirus disease 2019 (COVID-19) was mitigated by the introduction of efficient mass vaccination. However, severe acute respiratory syndrome coronavirus-2 (SARS-CoV-2) is still spreading worldwide, causing serious public health concerns and the need for new antiviral drugs, especially those that can be administered orally, is still an important issue. Other important weapons are monoclonal antibodies directed against SARS-CoV-2 S-protein, but they are only intravenously administrable and their activity could be reduced by the emergence of variants of concern (VOCs) [1].

Molnupiravir (EIDD-2801, MK-4482) and nirmatrelvir (PF-07321332) have recently been released for treatment of mild–moderate COVID-19 in frail patients, and they are both orally available. GC376 is a protease-inhibitor active against the feline coronavirus and it is often used as a reference for the evaluation of other potential inhibitors such as nirmatrelvir [2].

Molnupiravir is a new broad-spectrum antiviral drug, and, at the beginning of COVID-19 pandemic, it was in pre-clinical development for the treatment of influenza. It was originally designed for the treatment of alpha-virus infections and its target is the RNA-dependent RNA polymerase (RdRp) encoded by all RNA viruses. It is the most conserved RNA virus protein [3]. Molnupiravir is administered as an isopropylester pro-drug (EIDD-2801), which is rapidly cleaved in plasma to β-D-N_4_-hydroxycytidine (NHC; EIDD-1931). NCH, after distribution into tissues, is processed into NCH-triphosphate (MTP), the active metabolite of the drug [4]. Molnupiravir antiviral activity is based on the so-called “viral error catastrophe”; in particular, MTP can be used by the RdRp as a substrate instead of CTP or UTP, forming stable base pairs with a consequently proofreading escape. Subsequently, the mutated RNA products do not permit the formation of new complete viral particles. This two-step mechanism is probably the basis of the broad-spectrum antiviral activity of molnupiravir [5].

GC376 and nirmatrelvir are both peptidomimetics and inhibit coronaviruses 3-chymotrypsin-like cysteine (3CL) protease. These compounds are competitive inhibitors and share the pyrrolidone in the P1 position; however, nirmatrelvir uses a nitrile while GC376 uses an aldehyde to bind the catalytic cysteine. GC376, as for other first-generation inhibitors such as PF-00835231 and CDI-45205, has low oral bioavailability (3%). On the contrary, nirmatrelvir is a second generation 3CL-pro inhibitor with good oral bioavailability, the key element that allowed the development of this compound as an antiviral drug widely used in COVID-19 [2].

The lesson we have learned from HIV infection suggests that an antiviral combination could be more effective instead of a single drug, especially using different classes of drugs. Currently, protease inhibitors and nucleoside analogues are available for COVID-19 treatment. It is interesting to study in vitro and in vivo combinations of these classes, to establish if COVID-19 could benefit from antiviral combination treatment.

This study aims to investigate the in vitro activity of molnupiravir in combination with nirmatrelvir or GC376 on SARS-CoV-2 to verify their synergistic effect. The study was conducted on the wild-type virus and on the most widespread variants of concern (VOCs) during the current period, Omicron 1 and 2.

## 2. Materials and Methods

Methods are resumed in Appendix A.

### 2.1. SARS-CoV-2 Strains, Vero E6 Cell Cultures and Compounds

SARS-CoV-2 strains were isolated in the Biosafety Level 3 (BSL3) Virology Laboratory at “Santa Maria della Misericordia Hospital”, Perugia, Italy, as previously described [6] and used for all the experiments. The nasopharyngeal swab collected in transport medium (UTM) was incubated with a 1:1 nystatin (10,000 U/mL) and penicillin–streptomycin (10,000 U/mL) mixture for 1 h at 4 °C to remove bacterial/fungal contamination. After centrifugation at 400× *g* for 10 min, the supernatant was inoculated on an African green monkey kidney clone E6 (Vero E6) cells monolayer. Cells were maintained in Eagle’s minimum essential medium (EMEM) supplemented with 10% fetal bovine serum (FBS) and 1% penicillin–streptomycin at 37 °C with 5% CO_2_. The supernatant was withdrawn and titered by half-maximal tissue culture infectious dose (TCID50) endpoint dilution assay [7], aliquoted and stored at −80 °C.

Viral sequencing and variant assignation were performed at the Virology Laboratory of the Department of Medical Biotechnologies, University of Siena, Siena, Italy, as previously described [8]. Briefly, the sequencing of SARS-CoV-2 isolates was performed on the Illumina MiSeq instrument using the COVIDSeq Assay (Illumina) as indicated by manufacturer. The full-length viral genome was submitted to GISAID (http://gisaid.org/, accessed on 9 March 2022) to assign the correct variant.

Whole-genome sequencing of multiple isolates was used to identify a SARS-CoV-2 genome belonging to clade 20A.EU1 (lineage B.1) and clustered with viruses circulating in Europe from Spring to the end of 2020, and to Omicron sublineages (BA.1 and BA.2) [9]. The SARS-CoV-2 clade 20A.EU1 strain was isolated in May 2020 from a symptomatic patient during the first wave of infections. The Omicron variants were isolated in January and March 2022, respectively. Single virus stock aliquots were thawed immediately before each experiment and discarded after use.

Molnupiravir (EIDD-2801, Sigma-Aldrich, Merck, St. Louis, MO, USA), GC-376 (Selleck, Radnor, PA, USA) and Nirmatrelvir (PF-07321332, MedChemExpress, Monmouth Junction, NJ, USA) were suspended in DMSO at a concentration of 2 mg/mL following the manufacturer’s instructions. Stock aliquots were stored at −80 °C. For each experiment, the compounds were diluted to the desired concentration with EMEM supplemented with 10% of FBS.

### 2.2. SARS-CoV-2 Yield Reduction Assay and Cytotoxicity Assay

Vero E6 cells (3000 cells/well) were seeded in 96-well clear flat-bottom plates and incubated at 37 °C with 5% CO_2_ for 24 h. After incubation, cells were infected using a multiplicity of infection (MOI) of 0.1. SARS-CoV-2 was allowed to adsorb for one hour at 37 °C. Subsequently, virus inoculum was removed, and cells were overlaid with media containing 3-fold serial dilutions of molnupiravir (0.62–50 µM), nirmatrelvir (0.62–50 µM) and GC376 (0.21–16.7 µM). Negative controls (compounds alone), infected positive controls (SARS-CoV-2 alone) and mock-infected cells were included in each plate. Plates were incubated at 37 °C with 5% CO_2_ for 48 and 72 h and then, cell viability was measured using an MTT reduction assay [10].

After treatments, MTT (3-(4,5-Dimethyl-2-thiazolyl)-2,5-diphenyl-2H-tetrazolium bromide, Merck, Washington, DC, USA) solution (5 mg/mL) was diluted 1:10 with PBS and 100 μL were added to each well of the plates, which were subsequently incubated for 3 h at 37 °C with 5% CO_2_. Formazan crystals precipitated on the bottom of the wells were dissolved using 100 µL of DMSO incubated for 1 h. Absorbance at 570 nm with the reference filter at 630 nm was determined using a microplate reader (Tecan Infinite M200, Tecan Trading AG, Männedorf, Switzerland).

The percentages of cytotoxicity of each antiviral were calculated based on the respective vehicle (medium with DMSO) treated cells. The latter was used for the determination of the concentration able to inhibit cell growth by 50% as compared to the control (GI50). The selectivity index (SI), defined as the ratio of the 50% cytotoxic concentration (expressed as GI50) and the 50% effective concentration (EC50), was then calculated for each antiviral. A bioactive molecule with SI ≥ 10 is assumed to deserve further investigation.

Effective concentrations (EC50 and EC90) were calculated based on the analysis of the viability of infected cells by fitting drug dose–response curves using variable-slope regression modeling. Cell viability was assessed with MTT assay as described above.

### 2.3. Drug Combinations

The combinations were tested using 5 two-fold serial dilutions of each compound ≤EC90 as above described.

After 48 and 72 h post-infection, the supernatant was removed and stocked at −80 °C for further tests (plaque assay). Viability was performed using the MTT reduction assay as described above. Viability recovery was calculated as follows:$$Viability\;recovery\;\%=\frac{\left[treated\;infected\;cell\;viability-infected\;cell\;viability\right]}{\left[mock\;infected\;cell\;viability-infected\;cell\;viability\right]}\times 100$$

All tests were conducted in triplicate in three independent experiments.

To test whether the drug combinations act synergistically, the observed responses were compared with expected combination responses. The expected responses were calculated based on the highest single agent (HSA) reference model using SynergyFinder version 2 [11,12].

### 2.4. Plaque-Reduction Assay

After yield reduction assay, supernatant titers were determined by plaque assay as previously described [13]. Briefly, Vero E6 cells (300,000 cells/well) were previously seeded in a 12-well plate and incubated at 37 °C with 5% CO_2_ for 24 h. The medium was removed, and 250 µL/well of ten-fold serial dilution of supernatant were inoculated and incubated for 1 h. Plates were rocked every fifteen minutes with front-to-back and side-to-side movements. The inoculum was removed and 1 mL of overlay medium (complete medium with agar 0.1%) was poured into each well and the plates were incubated for 72 h. After incubation, the overlay was removed, and cells were fixed for 30 min with 4% formalin and stained with 0.5% crystal violet. Viral titer was determined as plaque-forming units per mL, considering wells with plaques ranging from 2 to 50. For each concentration of compound, viral titration was performed in triplicate.

### 2.5. Statistical Analysis and Data Elaboration

Statistical analysis was performed using GraphPad 8.3. Data were tested for normality using the Kolmogorov–Smirnov test and were presented as mean with the respective standard deviation (SD) or median with interquartile range (IQR), as appropriate. EC50, EC90 and EC99 concentrations were calculated using four-parameter variable-slope regression modeling. GI50 values were obtained using non-linear regression.

## 3. Results

Cytotoxicity assay of molnupiravir, GC376 and nirmatrelvir was performed on Vero E6 cells to obtain GI50 values. All the antivirals showed no significant cytotoxicity effects at the concentrations used in yield reduction assays with SI > 10 between antiviral and cytotoxic concentrations.

The antiviral effect of molnupiravir, nirmatrelvir and GC376 was then tested in vitro with Vero E6 cell-based viability test. Vero E6 cells were infected with SARS-CoV-2 20A.EU1 strain and then treated with different concentrations of antiviral drugs. After incubations, cell viability was determined through the MTT reduction assay. Four-parameter variable-slope regression modeling of molnupiravir dose–response after 48 h of treatment showed a half-maximal effective concentration (EC50) of 1.09 µM and an EC90 of 2.10 µM with a slope of 3.36 (95% confidence interval, CI 0.46–7.18) (Figure 1A). When the incubation was extended until 72 h, molnupiravir EC50 and EC90 were 1.24 and 3.74 µM, respectively, with a slope of 2.00 (95% CI 0.92 to 3.08) (Figure 1B).

Nirmatrelvir treatment for 48 h showed EC50 of 1.28 µM and EC90 of 3.70 µM with a slope of 2.07 (95% CI 0.51 to 9.00) (Figure 1A). After 72 h, nirmatrelvir EC50 and EC90 were 1.75 and 4.46 µM, respectively, with a slope of 2.35 (95% CI 0.37 to 5.07) (Figure 1B).

The protease inhibitor GC376 showed EC50 of 0.69 µM, and EC90 of 0.83 µM with a slope of 11.94 (range too wide for 95% CI calculation) after 48 h of incubation (Figure 1A). However, after 72 h of treatment, GC376 EC50 was 0.81 µM and EC90 was 1.85 µM with a slope of 2.68 (95% CI 1.30 to 6.66) (Figure 1B).

All the antivirals were found to be effective at micromolar/submicromolar concentrations. The subsequent experiments with combination treatment were performed using serial 2-fold dilutions starting from EC90 previously obtained. Vero E6 cells, infected with SARS-CoV-2 20A.EU1 strain, were treated with antiviral combinations and viability was determined as above.

Molnupiravir and nirmatrelvir showed a synergistic activity both at 48 and 72 h with an HSA score of 14.2 (*p* = 0.01) and 13.08 (*p* < 0.0001), respectively (Figure 2A,B).

On the other hand, molnupiravir and GC-376 showed a synergistic activity only at 48 h with an HSA score of 19.33 (*p* < 0.0001) and the interaction was additive at 72 h with an HSA score of 8.61 (*p* < 0.0001) (Figure 2C,D).

Three concentrations of cells treated for 72 h with antiviral combinations were selected to perform supernatant titration with plaque assay. For example, molnupiravir at a concentration of 0.94 µM showed a viability recovery of 54.9% compared to the positive control (virus + DMSO), while a viability recovery of 42.4% was found for nirmatrelvir at a concentration of 1.12 µM. The combination of the two compounds at these concentrations reached a viability recovery of 85.9%. Similar results were achieved with 0.47 and 1.87 µM molnupiravir concentrations combined with 0.57 and 2.23 µM nirmatrelvir concentrations, respectively.

As shown in Figure 3A, the combination of molnupiravir and nirmatrelvir reduced the viral titer significantly better than single agents (*p* = 0.02). The combination reduced the viral titer by an extra 0.4–2.1 log compared to molnupiravir alone (the more active compound at the concentrations tested). The same finding was obtained with the combination of molnupiravir and GC376 (Figure 3B, *p* = 0.02). The combination achieved an extra viral titer reduction of 0.4–1.1 log compared to molnupiravir alone.

Two concentrations of the combinations have been also tested on BA.1 and BA.2 (or Omicron 1 and 2) SARS-CoV-2 variants. For both variants, the combinations showed a trend of viral titer reduction but without reaching the statistical significance. As shown in Figure 4A, the molnupiravir and nirmatrelvir combination caused an extra viral titer reduction of 866.7–5000 PFU/mL (0.2–0.6 log) on the BA.1 strain compared to molnupiravir alone, but the differences were not significant (*p* = 0.1). Similar findings were shown by the molnupiravir–GC376 combination with a viral titer reduction of 783–3833.3 PFU/mL (0.1–0.5 log) more than molnupiravir alone (*p* = 0.1, Figure 4B). Omicron 2 strain showed similar behavior with both combinations: molnupiravir and nirmatrelvir at the higher concentration reduced viral titer by 103.3 PFU/mL (0.4 log) better than molnupiravir alone (*p* = 0.1) while the combination of molnupiravir with GC376 reduced the titer of 146.7–2166.7 PFU/mL (0.1–0.8 log, *p* = 0.1, Figure 4C,D).

## 4. Discussion

Different antiviral or potential antiviral drugs have been proposed for COVID-19 treatment based on in vitro tests and in silico studies. However, remdesivir, nirmatrelvir/ritonavir and molnupiravir are the only ones used in clinical practice for COVID-19 treatment. Considering the pandemic evolution, the availability of effective oral antiviral drugs that can be used at an early stage of SARS-CoV-2 infection is a priority in fighting COVID-19. Indeed, antiviral drugs play an important role to avoid the evolution into critical illness. Molnupiravir demonstrated a 30% reduction in hospitalizations and death in high-risk patients treated within 5 days of symptom onset. Nirmatrelvir/ritonavir reduced hospitalizations and death by 89% when administered within 3 or 5 days of symptom onset [14]. Other interesting compounds are now in development, such as the guanosine nucleotide analogue AT-527, the oral forms of remdesivir and GC376 [15,16,17].

Much of what we know about antiviral therapy is the result of HIV fight history. In particular, antiviral combination therapy was studied on this virus for the first time, showing that it is the only way to control HIV infection. Another important lesson can also be learned from the hepatitis C virus (HCV), in which the combination of direct-acting antivirals (DAAs) allowed the eradication of the infection irrespective of the viral genotype [18].

The great interest in the development of new antivirals does not end only with the COVID-19 pandemic. The pandemic has caused significant issues, not only in the health but also in the economic and social fields. The whole world did not have the resources to manage all the consequences of such a widespread spread of the virus. Therefore, COVID-19 demonstrated the importance to have plans and drugs to fight the next pandemics. Coronaviruses are not the only ones considered as potential high consequence viruses, but members of 11 virus families have been identified as potentially responsible for further pandemics [19]. In this perspective, the development of broad-spectrum antiviral drugs is mandatory. Some characteristics emerged as fundamental for an ideal antiviral therapy: the necessity of oral or inhaled drugs that could be taken at home and the use of antiviral combinations to improve potency and reduce resistance onset. An ideal cocktail would contain multiple agents targeting different coronavirus proteins, such as polymerase and proteinase [14].

Furthermore, the nucleoside/nucleotide analogues could benefit from the combination therapy with other compounds because it has been demonstrated that they are rapidly inactivated by the viral 3′-5′ exonuclease (nsp14), similar in all coronaviruses. This issue has been demonstrated for molnupiravir and remdesivir [20].

Starting from these considerations, this study was aimed to test antiviral combinations on SARS-CoV-2 and its widespread variants. We tested a polymerase inhibitor, molnupiravir, in combination with two proteinase inhibitors, nirmatrelvir and GC376. The effectiveness of these combinations could lay the foundation for the antiviral approach in the next coronavirus outbreaks. To the best of our knowledge, this is the first study that tested the combination of molnupiravir and GC376.

During the COVID-19 pandemic, different antiviral combinations have been studied and demonstrated synergic or additive activity, such as remdesivir with some HCV DAAs (i.e., pibrentasvir), nelfinavir or amodaquine, molnupiravir with nelfinavir, nirmatrelvir or favipiravir [12,14,20,21,22]. Interestingly, the antiviral drug cocktails of favipiravir with molnupiravir or remdesivir have also been tested in vivo in a SARS-CoV-2 Syrian hamster infection model, indicating reduction in both viral load and the histological lung pathology scores better than the single antiviral. The combination therapy also completely prevented transmission to co-housed untreated sentinels.

A wide number of molecules have been tested in combination against SARS-CoV-2, most of them in an in vitro model of Vero E6 cell lines and some in lung epithelial cells as Calu-3 [14].

Calu-3 cells are a more representative cell line because it better represents SARS-CoV-2 infection and pathogenesis: SARS-CoV-2 enters Vero E6 cells by fusion in endosomes, instead of lung cells that are infected by serine protease TMPRSS2-mediated cell surface fusion. If a compound tested on Vero E6 has a mechanism of action affecting endosomes, it could be not effective on epithelial cell lines and in vivo [14]. This is not the case for molnupiravir, nirmatrelvir and GC376 that, by inhibiting the polymerase and protease, respectively, could be efficiently tested on Vero E6. An example of this is the combination of favipiravir and molnupiravir that confirmed their activity in the Vero E6 model such as in vivo model [22].

Very few data are available in the literature about the combination of molnupiravir and nirmatrelvir. Li et al. demonstrated that the combination has an additive activity both in wild-type and Omicron SARS-CoV-2 strains [21]. We tested molnupiravir in combination with nirmatrelvir or GC376 using two different in vitro models. Firstly, we performed a synergistic checkerboard that was analyzed with a viability reduction assay. The effectiveness of the antiviral cocktails was tested after 48 and 72 h of incubation. It showed that both the combinations are synergic on the wild-type virus at 48 h of incubation. At 72 h, molnupiravir combined with nirmatrelvir showed synergistic activity while, when it was associated to GC376 an additive activity was found. Subsequently, we selected some significant concentrations of the two combinations and performed a plaque assay to confirm the results. We observed that the viability recovery found on the MTT assay corresponded to an effective viral titer reduction that was higher for the combination than for the single compounds.

We did not perform the synergistic checkerboard on Omicron 1 and 2 variants because we measured the antiviral effect as cell viability recovery, and it was not possible to perform this test on the Omicron 1 and 2 variants for their less efficient propagation on Vero E6. Of note, a similar propagation deficiency has also been observed on Calu-3 [21]. We tested some combination concentrations on Omicron 1 and 2 measuring the viral titer reduction. Both the combinations (molnupiravir plus nirmatrelvir and molnupiravir plus GC376) showed a trend of viral titer reduction better than the best compound alone but without reaching the statistical significance.

We also wondered about the possible synergistic mechanism of these antiviral cocktails. Nirmatrelvir and GC376 inhibit the protease 3CL that is responsible for the cleavage of 11 non-structural proteins including the exonuclease nsp14 that can excise molnupiravir from the SARS-CoV-2 RNA, thus inactivating it. We suggest that, with less effective activity of the SARS-CoV-2 exonuclease, molnupiravir may work better. These suggestions need to be demonstrated with further experiments.

The limits of the study are the in vitro setting that could overestimate the antiviral potency, the lack of other less common VOCs, and the use of non-human cell lines.

Further studies are necessary to confirm the effectiveness of these combinations, especially in vivo and in human models.

## 5. Conclusions

Our study demonstrates that molnupiravir in combination with nirmatrelvir or GC376 has a synergistic activity on SARS-CoV-2 in vitro, thereby suggesting new potential combination therapies against SARS-CoV-2. The next goal could be to realize pre-clinical studies with these combinations of antiviral drugs and then translate our findings into clinical trials. These combinations may have a global impact, improving especially the protection of the frail population from severe COVID-19.

## Figures and Tables

**Figure 1 microorganisms-10-01475-f001:**
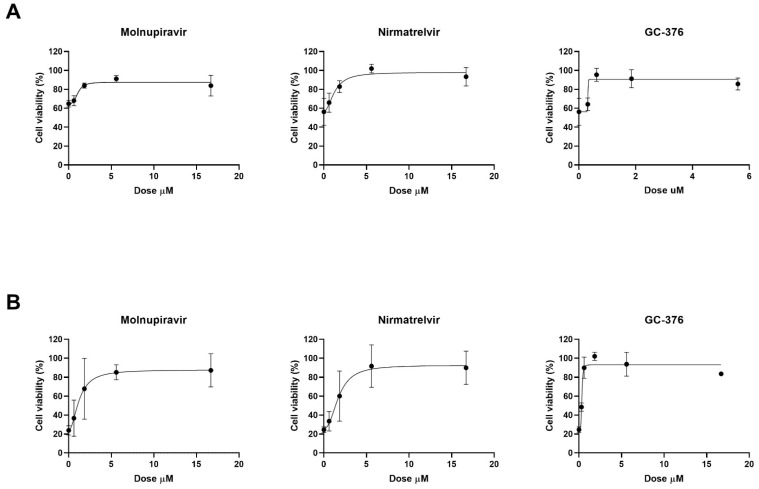
Four-parameter dose–response curve of molnupiravir, nirmatrelvir and GC376 dose–response. Vero E6 cells were infected with SARS-CoV-2 20A.EU1 strain and then treated with different concentrations of antivirals for (**A**) 48 and (**B**) 72 h. The viability of cells was assessed by MTT reduction assay and expressed as cell viability (%) ± SD. Data are the mean of at least 2 experiments with three technical replicates.

**Figure 2 microorganisms-10-01475-f002:**
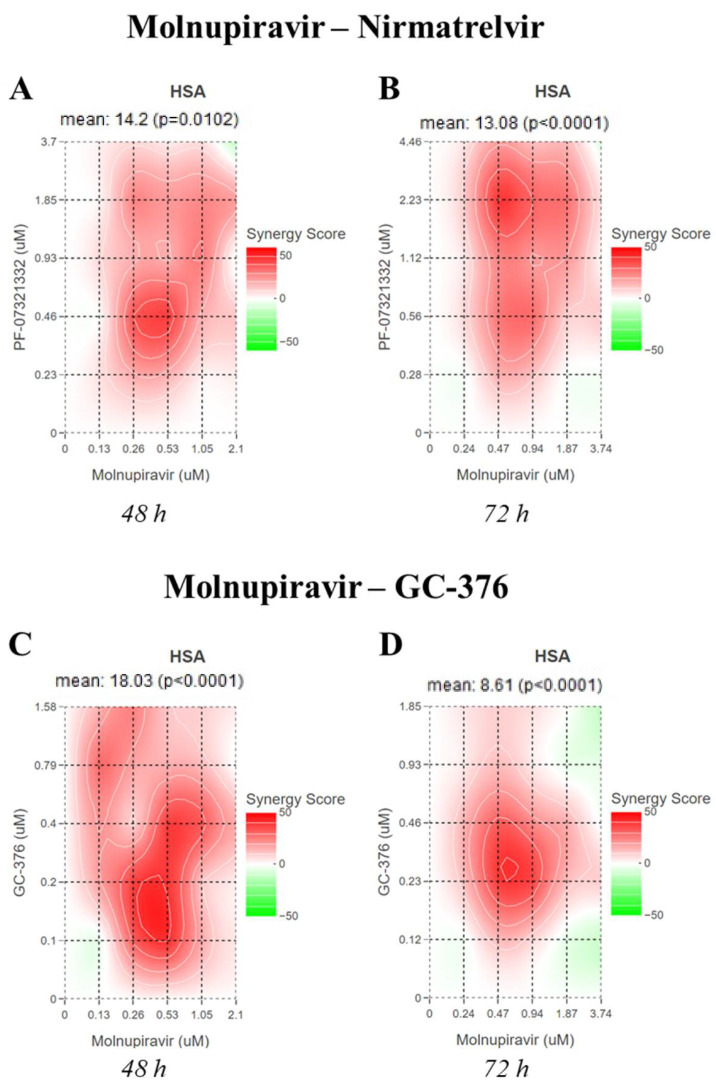
Synergy distribution in pairwise combination of antivirals. Vero E6 cell viability was determined after SARS-CoV-2 20A.EU1 strain infection and treatment with molnupiravir–nirmatrelvir combinations for (**A**) 48 and (**B**) 72 h, or the combinations of molnupiravir-GC376 for (**C**) 48 and (**D**) 72 h. Rescue of virus-mediated viability reduction was used for the analysis with the SynergyFinder version 2 tool. Data are from 3 independent experiments performed in triplicate.

**Figure 3 microorganisms-10-01475-f003:**
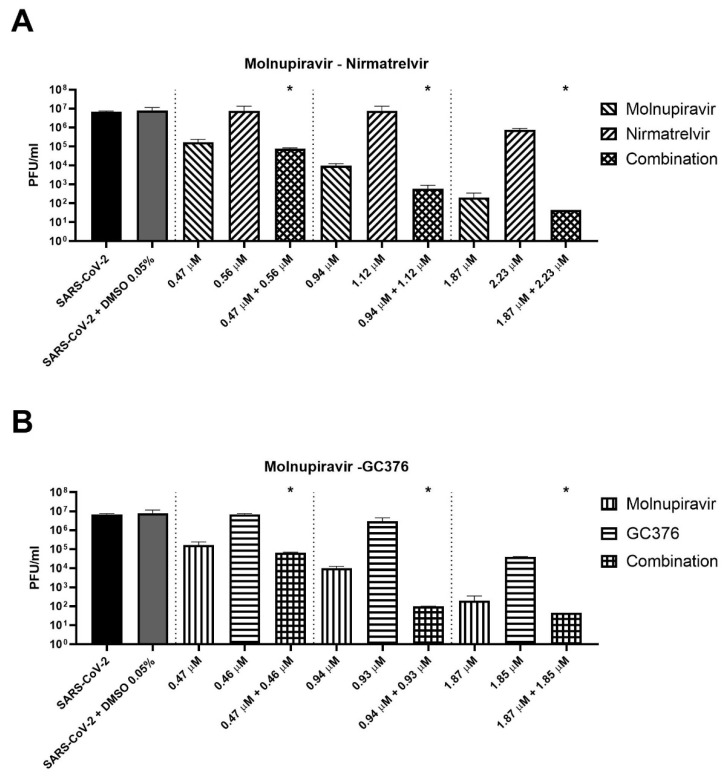
Plaque-reduction assay of the selected combination of antivirals. After synergy tests, supernatants of SARS-CoV-2 20A.EU1 infected Vero E6 cells treated with combinations of antivirals were frozen and selected concentrations of (**A**) molnupiravir–nirmatrelvir and (**B**) molnupiravir–GC376 were tested for viral load by plaque assay. Viral titers are expressed as mean ± SD of plaque-forming units (PFU)/mL from one experiment performed in triplicate. * *p* < 0.05, antiviral combination vs. more active single agent.

**Figure 4 microorganisms-10-01475-f004:**
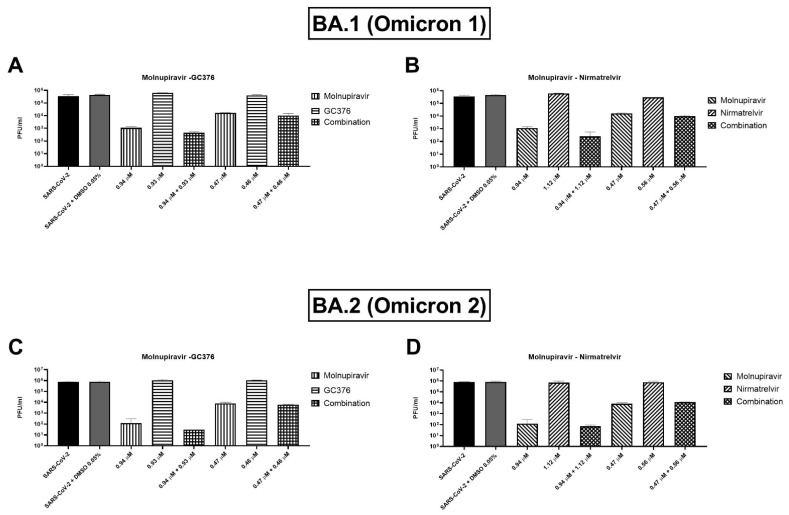
Plaque-reduction assay of the selected combination of antivirals. After synergy tests, supernatants of SARS-CoV-2 (**A**,**B**) BA.1 and (**C**,**D**) BA.2 infected Vero E6 cells treated with combinations of antivirals were frozen and selected concentrations of (**A**,**C**) molnupiravir–nirmatrelvir and (**B**,**D**) molnupiravir–GC376 were tested for viral load by plaque assay. Viral titers are expressed as mean ± SD of plaque-forming units (PFU)/mL from one experiment performed in triplicate.

## Data Availability

The datasets used and/or analyzed during the current study are available from the corresponding author on reasonable request.

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
