# Peer review of "The Combination of Molnupiravir with Nirmatrelvir or GC376 Has a Synergic Role in the Inhibition of SARS-CoV-2 Replication In Vitro"

_microorganisms, 2022, doi:10.3390/microorganisms10071475_

Round 1

Reviewer 1 Report

Comments:

1) Figure 1, 3: "Dose 'uM'" should be replaced with "Dose 'mu (Greek alphabet) M'".

2) Figure 1: Y axis label should be Cell viability (%) and 'OD570 nM' replaced with 'Cell viability (%)'.

3) Line 201: According to the figures (fig 1. A, B), all doses were not effective (same) at micromolar concentrations (for example 0.125 and 5.0 mkm shown different results). The sentence should be modified or removed.

4) Line 220-223: It is not necessary to show/describe (to compare with other combinations) about 0.47mkM (molnupiravir) and 0.56 mkM (nirmatrelvir) combination. Better to use 0.94 mkM+ 1.12 mkM because it shown better results (every where).

5) Line 226-227: This sentence should be modified for better understanding or described in details for each figures (3, 4).

6) Discussion should be modified (major) and need to discussed more about these two components and their combinations along with the research results obtained in this experiment.   

Author Response

Reviewer 1

1) Figure 1, 3: "Dose 'uM'" should be replaced with "Dose 'mu (Greek alphabet) M'".

Thank you for the correction.

2) Figure 1: Y axis label should be Cell viability (%) and 'OD570 nM' replaced with 'Cell viability (%)'.

Thank you for the suggestion, the figure has been modified.

3) Line 201: According to the figures (fig 1. A, B), all doses were not effective (same) at micromolar concentrations (for example 0.125 and 5.0 mkm shown different results). The sentence should be modified or removed.

The sentence has been modified.

4) Line 220-223: It is not necessary to show/describe (to compare with other combinations) about 0.47mkM (molnupiravir) and 0.56 mkM (nirmatrelvir) combination. Better to use 0.94 mkM+ 1.12 mkM because it shown better results (every where).

The text has been modified as you suggested.

5) Line 226-227: This sentence should be modified for better understanding or described in details for each figures (3, 4).

The sentence has been modified.

6) Discussion should be modified (major) and need to discussed more about these two components and their combinations along with the research results obtained in this experiment.  

Thank you for the suggestion. The discussion of results has been implemented.

Reviewer 2 Report

The manuscript of A. Gidari et al. “The combination of molnupiravir with nirmatrelvir or GC376 has a synergic role in the inhibition of SARS-CoV-2 replication in vitro” describes the effect of simultaneous application of anti-SARS-CoV-2 drugs on the virus’ reproduction in cell culture. Authors have demonstrated the synergistic effect of two compounds with different virus-specific targets and modes of action. The results obtained can be considered as experimental basis for further development and optimization of schedule(s) of COVID-19 treatment.

The set of methods chosen is adequate to the tasks of the study. The experiments are correctly performed, and the results clearly presented. The manuscript can be published in MDPI Microorganisms after minor corrections.

The Y axis on Fig.1 should be marked as “OD570 nm” instead of “OD570 nM” (nanometers, not nanomol).

The graphs on Fig.1 are not regression as indicated in the legend. The regression model suggests that higher dose of the drug (X) results in lower Y value. Would you please rename the graph type.

Author Response

Reviewer 2

The manuscript of A. Gidari et al. “The combination of molnupiravir with nirmatrelvir or GC376 has a synergic role in the inhibition of SARS-CoV-2 replication in vitro” describes the effect of simultaneous application of anti-SARS-CoV-2 drugs on the virus’ reproduction in cell culture. Authors have demonstrated the synergistic effect of two compounds with different virus-specific targets and modes of action. The results obtained can be considered as experimental basis for further development and optimization of schedule(s) of COVID-19 treatment.

The set of methods chosen is adequate to the tasks of the study. The experiments are correctly performed, and the results clearly presented. The manuscript can be published in MDPI Microorganisms after minor corrections.

The Y axis on Fig.1 should be marked as “OD570 nm” instead of “OD570 nM” (nanometers, not nanomol).

Thank you for the correction.

The graphs on Fig.1 are not regression as indicated in the legend. The regression model suggests that higher dose of the drug (X) results in lower Y value. Would you please rename the graph type.

The error has been fixed.

Reviewer 3 Report

I think this manuscript is good enough for the publication. English itself is well-written. Besides, everybody needs good information about drug treatments for COVID-19.

Author Response

Reviewer 3

I think this manuscript is good enough for the publication. English itself is well-written. Besides, everybody needs good information about drug treatments for COVID-19.

Thank you for the comment.

Reviewer 4 Report

Gidari et.al have conducted analysis of SARS-CoV-2 inhibitors (molnupiravir, nirmatrelvir and GC376 ) alone or in combination to test their efficacy, utilizing vero cell line. Their results show that combination of molnupiravir with nirmatrelvir or GC376 showed good additive-synergic activity in vitro. Monotherapy in the context of RNA viral infection is always a gamble, risking the development of resistance mutations, hence, dual or even triple therapy against SARS-CoV-2 infection deserves thorough investigation, however, results obtained by in vitro studies need to be confirmed in light of clinical trials.

The manuscript is concise and straight to the point, although, the methodology appears to be confusing, in addition to some points I would like to raise:

Abstract:

-        English language proofing is suggested to iron out mistakes such as „All experiments were conducted three times and in triplicate” in line 25, or “orally” administrable ones in line 39 ..etc.

Materials and methods:

-        2.2, what were the mock controls?

-        Why was absorbance measured at 570 after dissolving with DMSO and not 500-540 as usual? wouldn’t you get lower reads at this wavelength?

-        “Effective concentrations (EC50 and EC90) were calculated as the drug concentration able to improve cell viability by 50% and 90% compared to the infected control treated with the solvent alone (DMSO)” this definition is not accurate and confusing, also how was viability assessed?

-        Lines 137-145 is repetition, mentioned previously. I think the remaining 3 sentences do not warrant a dedicated sub-section and can be added to any other, also determination of serum concentration of the drugs was conducted by literature review, and does not belong in the Plaque assay section.

Results:

-        Figure 1. It would be more informative if the Y axis was changed to % viability relative to control, as is customary, instead of OD, and it should also be uniform in scale. Although, results would undoubtedly look better if plotted in compound graphs along with the inhibition assays but that is a matter of preference.

-        217-225 is not understandable in its current form, and it would be better to re-write it in a more comprehensible form. What is viability recovery? and what is it referring to?

-        Line 227: “better than the more active compound alone” both are active compounds

-        232-241: the results are indicated in log change, but the figures are represented by PFU/ml, I suggest unifying the two, or even plot as percentage of control, as it is very confusing in its current state and hard to follow. Additionally, if P value is not significant then you cannot assume reduction or otherwise.

-        Figure 3.: It would be better if the X-axis is in an increasing order of concentration, and maybe divide the concentration into separate groups to make the graph easier to read

-        Figure 4.: A,C, and B,D seem identical, even the controls. Given the different variants, and the nature of the assay, it seems odd to get same plaque numbers and SD

Discussion:

-        The discussion section needs thorough proof-reading for English language, and some parts require the use of better suited statements, such as lines259, 273-276, 289..etc.

Comments and Questions

-        The results obtained by the authors show that molnupiravir had an EC50 of around  0.9 uM, more or less within the known levels in the literature in vero cells, although slightly higher than anticipated (0,2-0,3 uM), however, it appears that nirmatrenvir was not effective (from Figure 3.A) at all. It is apparent from other similar studies that the IC50 is around 1 uM, utilizing similar experimental setup. How do the authors address this discrepancy?

-        My biggest concern with this manuscript is the methodology and the sequence of steps followed. According what I understood from the manuscript, viral samples were collected, TCID determined, from which MOI was calculated, and used to infect cells. Treatment with the inhibitors occurred after infection and the end point was calculating reduction in cytotoxicity ?

o   If so, as these are protease and RndP inhibitors, why weren’t the cells first treated and then infected? and how was cytotoxicity and “improvement” in cell viability assessed? plaque forming is unfortunately not an accurate model to assess efficacy, why not calculating viral copy numbers?

o   The authors need to clarify the methodology in order to make it easier to understand and assess

Author Response

Reviewer 4

Gidari et.al have conducted analysis of SARS-CoV-2 inhibitors (molnupiravir, nirmatrelvir and GC376 ) alone or in combination to test their efficacy, utilizing vero cell line. Their results show that combination of molnupiravir with nirmatrelvir or GC376 showed good additive-synergic activity in vitro. Monotherapy in the context of RNA viral infection is always a gamble, risking the development of resistance mutations, hence, dual or even triple therapy against SARS-CoV-2 infection deserves thorough investigation, however, results obtained by in vitro studies need to be confirmed in light of clinical trials.

The manuscript is concise and straight to the point, although, the methodology appears to be confusing, in addition to some points I would like to raise:

Abstract:

-        English language proofing is suggested to iron out mistakes such as „All experiments were conducted three times and in triplicate” in line 25, or “orally” administrable ones in line 39 ..etc.

The errors have been fixed.

Materials and methods:

-        2.2, what were the mock controls?

As specified in line 115, mock controls were included in each plate

-        Why was absorbance measured at 570 after dissolving with DMSO and not 500-540 as usual? wouldn’t you get lower reads at this wavelength?

We used a very low concentration of DMSO that did not affect the results. We measured absorbance at 570 nm because viable cells with active metabolism convert MTT into a purple colored formazan product with an absorbance maximum near 570 nm.

https://www.atcc.org/~/media/DA5285A1F52C414E864C966FD78C9A79.ashx

-        “Effective concentrations (EC50 and EC90) were calculated as the drug concentration able to improve cell viability by 50% and 90% compared to the infected control treated with the solvent alone (DMSO)” this definition is not accurate and confusing, also how was viability assessed?

Thank you for the suggestion. The sentence has been modified (lines 134-136).

-        Lines 137-145 is repetition, mentioned previously. I think the remaining 3 sentences do not warrant a dedicated sub-section and can be added to any other, also determination of serum concentration of the drugs was conducted by literature review, and does not belong in the Plaque assay section.

The repeated sentences have been deleted. The sub-section has been maintained because, as you suggested in another comment, we added a formula to better explain viability recovery calculation (lines 144-145).

The plaque assay section has been corrected, thank you for the suggestion.

Results:

-        Figure 1. It would be more informative if the Y axis was changed to % viability relative to control, as is customary, instead of OD, and it should also be uniform in scale. Although, results would undoubtedly look better if plotted in compound graphs along with the inhibition assays but that is a matter of preference.

As suggested, figure 1 has been modified.

-        217-225 is not understandable in its current form, and it would be better to re-write it in a more comprehensible form. What is viability recovery? and what is it referring to?

Thank you for the suggestion. The sentences have been simplified. We added a formula to better explain viability recovery calculation (lines 144-145).

-        Line 227: “better than the more active compound alone” both are active compounds

Thank you for the suggestion, the sentence has been modified.

-        232-241: the results are indicated in log change, but the figures are represented by PFU/ml, I suggest unifying the two, or even plot as percentage of control, as it is very confusing in its current state and hard to follow. Additionally, if P value is not significant then you cannot assume reduction or otherwise.

Thank you for the suggestion. We added the viral titer reduction in PFU/ml. We also further specified that the reductions were not significant.

-        Figure 3.: It would be better if the X-axis is in an increasing order of concentration, and maybe divide the concentration into separate groups to make the graph easier to read

Thank you for the suggestion, the figure has been modified.

-        Figure 4.: A,C, and B,D seem identical, even the controls. Given the different variants, and the nature of the assay, it seems odd to get same plaque numbers and SD

The error has been fixed.

Discussion:

-        The discussion section needs thorough proof-reading for English language, and some parts require the use of better suited statements, such as lines259, 273-276, 289..etc.

Thank you for the suggestion. The discussion section has been modified.

Comments and Questions

-        The results obtained by the authors show that molnupiravir had an EC50 of around  0.9 uM, more or less within the known levels in the literature in vero cells, although slightly higher than anticipated (0,2-0,3 uM), however, it appears that nirmatrenvir was not effective (from Figure 3.A) at all. It is apparent from other similar studies that the IC50 is around 1 uM, utilizing similar experimental setup. How do the authors address this discrepancy?

Data obtained from cell viability tests showed an IC50 of 1.28 uM at 48 h and 1.75 uM at 72 h for Nirmatrelvir. The plaque assay tests showed that the concentration of 2.23 uM reduced the viral titer by 90%. Furthermore, analysing data shown in Figure 3 (plaque assay) through a four-parameter variable-slope regression modelling we found an IC50 of 1.65 uM for Nirmatrelvir (data not shown), not so far from the finding mentioned above.

-        My biggest concern with this manuscript is the methodology and the sequence of steps followed. According what I understood from the manuscript, viral samples were collected, TCID determined, from which MOI was calculated, and used to infect cells. Treatment with the inhibitors occurred after infection and the end point was calculating reduction in cytotoxicity ?

Thank you for the question. Everything is correct. The effect was calculated in the reduction of the cytopathic effect. We added a flowchart to make the understanding of the method easier (Supplementary Figure 1).

o   If so, as these are protease and RndP inhibitors, why weren’t the cells first treated and then infected?

The cytopathic effect usually appears some hours after viral infection. We treated cells with the compound after 1 h of infection, the estimated time for viral cell entry. Some studies used similar methods (for example DOI: 10.1038/s41422-022-00618-w).

and how was cytotoxicity and “improvement” in cell viability assessed?

Cytotoxicity of compounds was measured using an MTT reduction assay (lines 117-118). We added a formula to better explain viability recovery calculation (lines 144-145).

 plaque forming is unfortunately not an accurate model to assess efficacy, why not calculating viral copy numbers?

Thanks for the suggestion. The main method we used to measure the effectiveness of the compounds was the viability test. Plaque assay was used to confirm the efficacy in determining the release of infecting viral particles on the supernatant. Certainly, viral copy numbers count could be a valid alternative, but we showed that the two different methods used brought concordant findings. Furthermore, other articles used only cell viability and plaque assay for similar experiments (doi:10.3390/v12101178; the citation of this article has been added, see line 150).

o   The authors need to clarify the methodology in order to make it easier to understand and assess

As suggested, in order to clarify the methodology a flowchart and the formula used to calculate viability recovery have been added

Round 2

Reviewer 4 Report

The authors have implemented the suggested changes and the supplementary figure is very beneficial. The manuscript has been improved. I can only recommend that the authors proof-read their manuscript to iron out grammatical mistakes and improve the Enlish language especially in the discussion section.

For example:

Line 25: All experiments were conducted in triplicates is sufficient, as triplicate means three times.

Line 81: Outline of the methodology is indicated in supplementary Figure 1.

Line 146: as described above

etc.

This manuscript is a resubmission of an earlier submission. The following is a list of the peer review reports and author responses from that submission.